# Healthcare Costs of Hospitalizations Due to Aspergillosis and 25-Year Trends in Spain, 1997–2021

**DOI:** 10.3390/jof10110733

**Published:** 2024-10-23

**Authors:** María Rincón Villar, Montserrat Alonso-Sardón, Elisa Alvarez-Artero, Beatriz Rodríguez Alonso, Amparo López-Bernús, Ángela Romero-Alegría, Javier Pardo-Lledías, Moncef Belhassen-García

**Affiliations:** 1Servicio de Medicina Interna, Complejo Asistencial Universitario de Palencia (CAUPA), Av. Donantes de Sangre, 34005 Palencia, Spain; mrinconv@saludcastillayleon.es; 2Área de Medicina Preventiva y Salud Pública, Instituto de Investigación Biomédica de Salamanca (IBSAL), Centro de Investigación en Enfermedades Tropicales (CIETUS), Universidad de Salamanca, 37007 Salamanca, Spain; sardonm@usal.es; 3Servicio de Medicina Interna, Unidad de Infecciosas, Complejo Asistencial Universitario de Salamanca (CAUSA), Instituto de Investigación Biomédica de Salamanca (IBSAL), Centro de Investigación en Enfermedades Tropicales (CIETUS), Universidad de Salamanca, 37007 Salamanca, Spain; beamedicina@gmail.com (B.R.A.); aralegria@yahoo.es (Á.R.-A.); 4Servicio de Medicina Interna, Sección de Enfermedades Infecciosas, Complejo Asistencial Universitario de Salamanca (CAUSA), Instituto de Investigación Biomédica de Salamanca (IBSAL), Centro de Investigación en Enfermedades Tropicales (CIETUS), Universidad de Salamanca, Paseo San Vicente 58-182, 37007 Salamanca, Spain; alopezb@saludcastillayleon.es; 5Servicio de Medicina Interna, Hospital Marqués de Valdecilla, IDIVAL (Instituto de Investigación Valdecilla), Universidad de Cantabria, 39008 Santander, Spain; javipard2@hotmail.com

**Keywords:** aspergillosis, human aspergillosis, invasive fungal diseases, economic, healthcare costs, Spain

## Abstract

In the last 40 years, a significant increase in the incidence of lung infections by Aspergillus has been reported. The scarcity of studies that describe the costs of aspergillosis indicates that the economic impact of aspergillosis in the hospital environment is greater than that of other fungal infections. The objective of the study was to evaluate the direct healthcare costs associated with aspergillosis in the Spanish National Health System from 1997 to 2021. A retrospective nationwide longitudinal descriptive study was designed to review hospital records from the Minimum Basic Data Set of patients admitted to hospitals of the National Health System from 1997 to 2021, with a diagnosis of aspergillosis. A total of 44,586 patients were admitted for aspergillosis in the Spanish National Health System. There was a progressive increase in the average annual cost from 1997 to 2012, which reached a maximum peak, EUR 1,395,154.21 (±2,155,192.87). It decreased between 2014 and 2019, but increased again in 2020 and 2021, EUR 28,675.79 (±30,384.12). The Pearson correlation coefficient revealed a weak negative correlation between age and hospital costs and a moderate positive correlation between average length of stay and hospital costs. Our data show that the economic impact of hospitalizations for aspergillosis is significant and increasing at a rate proportionally higher than that of other prevalent diseases. Costs related to Aspergillus infection are associated mainly with respiratory diseases. The results of this economic evaluation may be useful for health authorities to develop a future economic strategy for managing this fungal infection.

## 1. Introduction

Aspergillosis is caused by the species of the fungus *Aspergillus* spp., a common saprophytic colonial fungus that is ubiquitously found in nature and affects susceptible hosts when inhaled. The species most frequently involved are *A. fumigatus*, *A. flavus*, *A. niger*, and *A. terreus* [1,2]. Depending on the underlying immune status of the host, the fungus may cause disease through any of the following mechanisms: fungal sensitization, allergic response, saprophytic colonization, and/or frank invasion of the lung parenchyma. Hence, it is now regarded as a semicontinuous spectrum of allergic, noninvasive, and invasive forms [3].

Aspergillosis is described mainly in neutropenic patients due to hemato-oncological diseases [1,4,5]. The lethality rate has decreased significantly in recent years owing to early treatment, especially in invasive aspergillosis, where a decrease of up to 30% has been observed [6]. In severely immunosuppressed patients, the lethality rate reaches up to 40–50% [1,7]. In recent years, an increasing number of cases has been reported, especially of invasive aspergillosis [2,7]. This is mainly due to the group of patients receiving immunosuppressive treatments and the increase in the number of solid organ transplants, with an incidence of up to 19% in this group [7].

Since 2013, the Leading International Fungal Education (LIFE) portal estimated the burden of aspergillosis infections by country, revealing differences in the global burden between countries, within regions of the same country, and between at-risk populations [8]. Therefore, according to epidemiological reports, a significant increase in the incidence in the last 40 years likely reflects an increase in the number of immunocompromised patients [7,9].

However, very little information is available concerning the overall hospital cost of fungal infections caused by *Aspergillus* spp. [10]. The scarcity of studies that describe the costs associated with aspergillosis indicates that the economic impact of aspergillosis in the hospital environment is greater than that of other fungal infections [9,11,12]. This economic cost has usually been analyzed in a partial and incomplete manner. Thus, most works focused on invasive aspergillosis and have fundamentally evaluated the cost associated with changes in therapeutic strategy between the different families of antifungals in countries such as the United States, Canada, Sweden, and Spain [12,13,14,15]. Consequently, there are currently few studies on the economic impact of aspergillosis, and comparisons between countries are difficult. Therefore, the aim of our study was to evaluate the direct healthcare costs associated with aspergillosis among inpatients in the Spanish National Health System (NHS) from 1997 to 2021.

## 2. Materials and Methods

### 2.1. Study Population and Data Source

We conducted a nationwide retrospective longitudinal descriptive study of all aspergillosis hospitalizations in public hospitals of the Spanish NHS between 1 January 1997 and 31 December 2021. Spain’s NHS provides publicly funded universal coverage and offers a broad portfolio of services that includes all technologies and health procedures for known diseases. It comprises all the health services of the state administration and the 17 autonomous communities that constitute the Spanish state.

The necessary information was obtained from the Minimum Basic Data Set (MBDS/CMBD in Spanish), from its first to its latest version, which is called the Register of Specialized Health Care (RAE-CMBD in Spanish). The latest version was implemented in 2016 and extended the register to other forms of in-patient care (e.g., home care, medical day hospital, out-patient surgery, out-patient procedures of special complexity, and emergencies) and to the private sector. The CMBD is a specialized healthcare registry that uses each care contact with a patient as a recording unit. It is an essential source for epidemiological research in the NHS, which has existed for more than 25 years. Until 2015, Spain applied the International Classification of Diseases (ICD)-9-CM (Clinical Modification of the 9th Revision of the International Classification of Diseases) as the reference classification for the coding of diagnoses and procedures of the CMBD. Since 1 January 2016, the ICD-10 has been implemented as the reference classification for clinical coding and recording of morbidity, in accordance with the Spanish Royal Decree 69/2015 [16].

The principal diagnosis is defined as the condition after the study that results in admission to the hospital, according to the ICD-9-CM or ICD-10 Official Guidelines for Coding and Reporting and the criteria of the attending clinical service or medical practitioner, even if major complications or other independent conditions have occurred during the stay. Secondary diagnoses are “other diagnoses” or conditions that coexist with the principal condition at the time of admission (i.e., comorbidities) or develop later during the hospital stay (i.e., complications), influencing the length of stay or the treatment administered.

The classification of Spanish public hospitals into clusters establishes the following categories. GROUP 1: Small regional hospitals with fewer than 150 beds on average, with hardly any high-tech equipment, few doctors and low complexity of care. GROUP 2: Basic general hospitals, average size of less than 200 beds, minimal technological equipment, with some teaching weight and somewhat greater complexity of care. GROUP 3: Area hospitals, average size of approximately 500 beds. On average, they have more than 50 MIR (Medical Intern Resident) doctors and 269 doctors. Medium complexity (1.5 complex services and 1.01 case mix). GROUP 4: Group of large hospitals, but more heterogeneous in staffing, size, and activity. High teaching intensity (more than 160 MIR doctors and high complexity (four complex services on average and a case mix greater than 1.20)). GROUP 5: Hospitals of great structural weight and high activity. Full range of services. More than 680 doctors and approximately 300 MIRs were included. These include large complexes. GROUP 6: Hospitals that could not be classified or that had not been assigned to any of the previous groups at the time of analysis.

The health administration uses the “Diagnosis Related Groups” (DRGs) classification system to define hospital costs [16]. This system groups patients with various diagnoses of similar resource consumption into a case mix category. Each patient is assigned a single DRG at discharge. The Ministry of Health periodically prepares these costs of hospitalization processes. The DRG classification used in Spain until 2015 was the All Patient DRG (AP-DRGs), which was designed exclusively for clinical variables to be coded with the ICD-9-CM. Since changing to the ICD-10, AP-DRGs have ceased to be used worldwide and have been replaced by All Patients Refined (APR-DRGs). Each APR-DRG is classified into four types of severity of illness (SOI) and four types of risk of mortality (ROM) subclasses, ranging from 1 to 4 (1-minor, 2-moderate, 3-major, and 4-extreme), which are determined by secondary diagnoses, with weights increasing as severity increases.

The estimation of the patient costs is an approximate calculation based on the DRG classification system. Each condition treated is assigned a code from the DRG classification system. Each of these DRGs has an average weight (standard value in each of the hospitals of the NHS that allows the comparison of activity between hospitals). Each AP-DRG or APR-DRG (and with each severity level since 2016) has its associated weights and costs. The cost is calculated by multiplying the number of cases of each DRG (and severity level since 2016) by its average cost and dividing by the total number of cases of a given unit, group, or process. Until 2015, the costs corresponded to the costs of the AP-DRG estimated in the process of estimating hospital costs of the NHS, so that each discharge was assigned the estimated cost for the AP-DRG that corresponds to it. Since 2016, the average cost has been different for each DRG and severity level.

### 2.2. Selection of Aspergillosis

We included all aspergillosis hospitalizations in public hospitals of the NHS between 1997 and 2021. We defined an aspergillosis-related hospitalization using the classifications of the International Classification of Diseases, 9th Revision, Clinical Modification (ICD-9-CM) and 10th edition (ICD-10) as follows: any hospital discharge with a principal or secondary diagnosis of aspergillosis and infection due to Aspergillus species (semicontinuous spectrum of allergic, noninvasive, and invasive forms, chronic pulmonary aspergillosis included) diagnosis codes: 117.3 (ICD-9-CM, cases 1997–2015) and B44 (ICD-10, cases 2016–2021). The CMBD excludes short-term admissions (<24 h), emergency department care, and outpatient care. Patients with missing data were excluded from the study.

### 2.3. Data Analysis

The total cost of aspergillosis-related hospital care was estimated as the sum of the total hospital charges incurred for all hospitalizations for which aspergillosis was listed as a diagnosis. The average cost corresponds to the average cost estimated for each DRG in the process of estimating the weights and NHS costs of the version in force, which are calculated and updated for the reference year. For the APR-DRG, the cost is calculated for each severity level (SOI). The methodological difference between AP-DRGs and APR-DRGs in cost calculation required us to estimate the costs of aspergillosis-related hospital care separately in the two following periods: from 1997 to 2015 and from 2016 to 2021. All costs were expressed in euros (EUR).

An initial descriptive analysis of the variables was conducted. Continuous variables were expressed as the means and standard deviations (SDs) or medians, interquartile ranges (IQRs), and ranges. Categorical variables were reported in absolute values (n) and percentages (%). The means of the quantitative variables and proportions of the qualitative variables were subsequently compared. Qualitative variables were tested with Pearson’s chi-square test to assess trends. ANOVA was used to determine the existence of statistically significant differences between two or more categorical groups by testing for differences in means using one variance. The Pearson correlation coefficient (r) was used to describe the strength and direction of the linear relationship between two quantitative variables. Statistically significant differences were those with *p* < 0.05. The statistical analysis was performed via the statistical program SPSS 28.0.

### 2.4. Ethics Statement

The study protocol was approved by the Clinical Research Ethics Committee of Investigation with Drugs of Cantabria, Spain (CEIMC 2020.353). The procedures described here were carried out in accordance with the ethical standards described in the Revised Declaration of Helsinki in 2013.

## 3. Results

### 3.1. Patient Characteristics

A total of 44,586 patients were admitted for aspergillosis in the Spanish National Health System between January 1997 and December 2021; in most cases (82%), the diagnosis of Aspergillus was secondary. A total of 30,023 (67%) were men, with a mean age of 63 (±17.6) years of age, compared to 14,560 women (33%), with a slightly lower mean age of 59 (±20.7) years of age (*p* < 0.001); 1347 (3%) were in the pediatric population. Hospital admissions were mainly non-programmed (77.5%), more frequent during the winter season (31.2%), and in specialized hospitals, such as area hospitals (GROUP 3, 25.7%), large hospitals (GROUP 4, 21.6%), and complex hospitals (GROUP 5, 26.6%) of the Spanish public health system (97%). Aspergillus infection was mainly associated with respiratory diseases such as chronic obstructive pulmonary disease (COPD) (44%), influenza/pneumonia (32.4%), and hematological malignancies (20.4%). The hospital services with the highest number of admissions were internal medicine (20.8%), pneumology (19.7%), clinical hematology (13.5%), and medical processes (81.1%). A total of 3512 (7.9%) patients needed a critical care unit (CCU)/intensive care unit (ICU), and 68 (0.1%) needed a neonatal intensive care unit (NICU)/pediatric intensive care unit (PICU). Furthermore, 650 (1.5%) patients were at the transplant unit. The mean hospital stay was 27 (±27.3) days. Most patients were classified as having SOI or ROM levels of 3 (major) or 4 (extreme). Finally, one out of four patients died (25%). Table 1 shows the clinical and epidemiological characteristics of these patients.

### 3.2. Cost of Aspergillosis-Related Hospitalizations

There was a progressive increase in the average annual cost from 1997 to 2012, which peaked at EUR 1,395,154.21 (±2,155,192.87). It decreased between 2014 and 2019 but increased again in 2020 and 2021, reaching EUR 28,675.79 (±30,384.12). Figure 1 shows the evolution of the average cost (EUR) year by year in Spain over the 25 years of the study, 1997–2021. Similarly, Figure 2 shows the evolution of the average cost (EUR) year by year in the different Spanish hospital clusters over the last 25 years. The average cost increases as the cluster increases in complexity and specialization.

The average and total costs associated with Aspergillus infection in Spain over this 25-year period are shown in Table 2. We highlight some total costs as follows: in special units like ICU costs were EUR 4,152,2570,859 (Mean cost per patient: 1997–2015 was EUR 2,572,988 vs. 2016–2021 was EUR 38,485); regarding comorbidity, COPD costs were EUR 8,817,805,808 (Mean cost per patient: 1997–2015 was EUR 691,676 vs. 2016–2021 was EUR 10,933), influenza costs were EUR 11,597,012,286 (Mean cost per patient:1997–2015 was 1,225,153 vs. 2016–2021 27,644) and Hematology malignancy EUR 8,868,538,550 (mean cost per patient: 1997–2015 was EUR 1,377,734 vs. 2016–2021 EUR 19,556). Regarding outcomes, total costs of survival was EUR 18,732,760,191 (mean cost per patient:1997–2015 was EUR 869,866 vs. 2016–2021 EUR 14,074) and total costs in death was EUR 10,442,003,681 (mean cost per patient: 1997–2015 was EUR 1,443,287 vs. 2016–2021 EUR 28,091). An ANOVA revealed significant differences in the costs related to Aspergillus infection for all the variables analyzed (*p* < 0.001). The Pearson correlation coefficient indicated a weak negative correlation between age and hospital costs (r = −0.144, period 1997–2015; r = −0.107, period 2016–2021), with lower ages resulting in higher hospital costs, and a moderate positive correlation between average length of hospital stay and hospital costs, with hospital costs increasing as the number of days of hospital stay increase (r = 0.389, period 1997–2015; r = 0.503, period 2016–2021). Finally, Figure 3 illustrates the costs associated with aspergillosis in Spain over a 25-year period, both for patients who survived and for those who died.

## 4. Discussion

We evaluated the economic impact of hospitalized patients with aspergillosis in the Spanish National Health System from 1997 to 2021. Our data revealed that the economic impact of hospitalization for aspergillosis is significant and increasing. Notably, other diseases that are more prevalent than aspergillosis in Spain, such as chronic heart failure (CHF), diabetes (DM), or community-acquired pneumonia (CAP), have a proportionally lower economic impact than the pathology studied. For example, CHF is estimated to cost EUR 2500 million per year, of which 470 million are exclusively hospital costs. In 2010, DM generated a cost of EUR 5809 million in Spain. Finally, at the European level, CAP is estimated to cost EUR 10.1 billion, with a hospital burden of EUR 5.7 billion, and at the national level, it is estimated to generate costs of EUR 5353/patient.

Previous studies and publications by our group have shown an increase in the incidence of aspergillosis in Spain [2,17], which leads to the associated economic costs. Other reasons include the increase in immunocompromised patients and the prolongation of survival in these patients [2,17].

Since 1 January 2016, the ICD-10-ES classification has been the reference of classification for clinical coding and morbidity registration in Spain, replacing the ICD-9-CM. The quality of the ICD-10 is higher than that of the ICD-9 because of the improved specificity of the codes. Better coding means better quality hospital data, which translates into better analysis, better measurement of services provided and, consequently, a more accurate and detailed costing methodology. This, among other factors, could explain the increase in cases.

We note the great difference between the expenses incurred by aspergillosis and those incurred by other services. Thus, hematological patients double the expenses generated (mean cost per patient) in other services, such as pulmonology or internal medicine. This can be explained by what was described by the American Society of Microbiology and the Spanish Public Health Society, which observed a greater number of complications and coinfections due to the state of immunosuppression in hematological patients [1,7,18,19,20]. Similarly, pediatric patients have higher cost figures, possibly because they are onco-pediatric patients.

During the 25-year study period, two peaks in hospital costs were observed. The first was between 2010 and 2012, which could be related to the first influenza A pandemic [21]. The second peak between 2020 and 2021 coincided with the SARS-CoV-2 pandemic [2,17]. Our study supports the hypothesis already described of the association between infection by *Aspergillus* spp. and different viruses, such as influenza and, more recently, SARS-CoV-2. Reinforcing the idea that coinfection usually entails a greater use of resources, the need for intensive care, ventilatory support therapies, a greater number of days of hospitalization, and greater lethality, it ultimately generates a greater economic cost [22,23,24,25]. Notably, between 2015 and 2018, a decrease in expenses was observed. One of the factors that has favored economic savings may be the therapeutic introduction of isavuconazole compared with other more expensive options, a phenomenon already supported in several studies in the United States, Europe, and Spain [12,13,14,15]. Isavuconazole was approved in March 2015 by the Food and Drug Administration (FDA) and was marketed in Spain in 2017.

We wanted to highlight the relationship between the type of hospital and the economic impact caused by aspergillosis. Thus, the more complex the hospital is, the greater the expense. It seems logical to think that these hospitals, many of which are reference hospitals, manage the most complex patients. In addition, disease severity (SOI) and mortality risk (ROM) are mainly determined by the interaction of multiple diseases. Thus, patients with greater disease severity had higher costs.

The APR-DRG system is based mainly on secondary diagnoses instead of classification by age and complications or comorbidities, such as the previous version of AP-DRG; therefore, patients with a greater number of secondary diagnoses (and therefore greater SOI) consume a greater amount of health resources, which translates into higher costs, and these data are compatible with previous Spanish studies [26].

Variations in the distribution of severity are directly related to variations in the average weight for these DRGs in APR (since 2016), whereas in the AP (until 2015), this weight is constant. With the implementation of the ICD-10 for diseases and procedures, the grouping was revamped, including the introduction of a refined grouping for all patients (APR-DRG), which was stratified into four levels of severity (according to resource consumption) that were mainly determined by secondary diagnoses reflecting the comorbidities and severity of the principal diagnosis, with weights increasing with increasing severity. The introduction of the ICD-10 and APR-DRG necessitated a better and more exhaustive definition of processes, demanded a higher quality of coding than previous versions (measured as the average number of diagnoses and procedures per episode), and revealed some inconsistencies in the equivalences for some procedures that have been used until then.

The use of the APR-DRG reduces the complexity (mean weight) of a hospital by half compared with that obtained if the same episodes are grouped with the AP-DRG. However, the distribution according to the four levels of severity in the APR-DRG is highly variable across hospitals. The impact of robust information recording on complexity is much greater in the APR-DRG than in the AP-DRG. This leads us to affirm that the impact of a good recording of information on complexity or average weight is greater in the APR-DRG than in the AP-DRG [26,27].

### Strengths and Limitations

The main limitation of this work is that it is a retrospective study, and the information used comes from the CMBD. The CMBD provides information from a network of hospitals that covers more than 99% of the population living in Spain (https://www.sanidad.gob.es accessed on 2 March 2024) thus, we are confident that this study provides fairly accurate estimates. However, several factors contributed to the limitations of our study as follows: (i) the use of sources, such as the CMBD, for purposes other than research and clinical care; (ii) the use of the ICD-9 code, which has certain classification limitations with respect to the ICD-10; (iii) encoding errors; (iv) the inability to access patient medical history, which prevented us from confirming the diagnosis and the species implicated, identifying the possible associated factors involved, and assessing the tests used for aspergillosis diagnosis, which impaired the quality of the data. To our knowledge, there are no data on the sensitivity and specificity of the 117.3 code (1997–2015) and code B44 (2016–2017) for aspergillosis cases. (v) This study only considered cases in hospitals and not non-hospital cases or those in private centers; for example, patients who are ill and who are not admitted or who did not receive medical care, in addition to those treated in private hospitals, were excluded. Thus, hospital records underestimate the real burden of aspergillosis in Spain. (vi) The DRGs were not designed to be used for health services research, though they have proven useful for this purpose. The DRG cost databases attach monetary value to hospital healthcare resources consumed by patients. Their advantages include public and free access, transparency, exhaustivity (accounting for severity and risk levels), regular updates, and a classification system that is stable over time and comparable across hospitals and even countries [28]. Both the DRG costs and DRG weights are published by the Spanish NHS. The cost calculation methodology was revised in 2016. However, Spain has a decentralized public administration system, with 17 autonomous communities (ACs) and two autonomous cities (Ceuta and Melilla), each with different hospital accounting procedures and practices. Thus, some ACs use top-down gross costing methods that are based on North American DRG weightings to allocate costs to DRGs, whereas other ACs use micro-costing to determine cost drivers. Public tariffs/prices for inpatient DRGs are published separately for the 17 ACs and two ACs [29]. Furthermore, the APR-DRG accounts for diagnoses and procedures, but not complications that may occur during hospitalization and may impact the overall cost [30]. We acknowledge these limitations but also believe that we have contributed to the generation of hypotheses that may be explored in further investigations.

## 5. Conclusions

In conclusion, our data show that the economic impact of hospitalization for aspergillosis is significant and increasing at a rate proportionally higher than that of other diseases, such as chronic heart failure, diabetes, or community-acquired pneumonia. Costs associated with Aspergillus infections were mainly associated with respiratory diseases such as chronic obstructive pulmonary disease or influenza/pneumonia and hematological malignancies. The results of this economic evaluation may be useful for health authorities to develop a future economic strategy for the management of this fungal infection.

## Figures and Tables

**Figure 1 jof-10-00733-f001:**
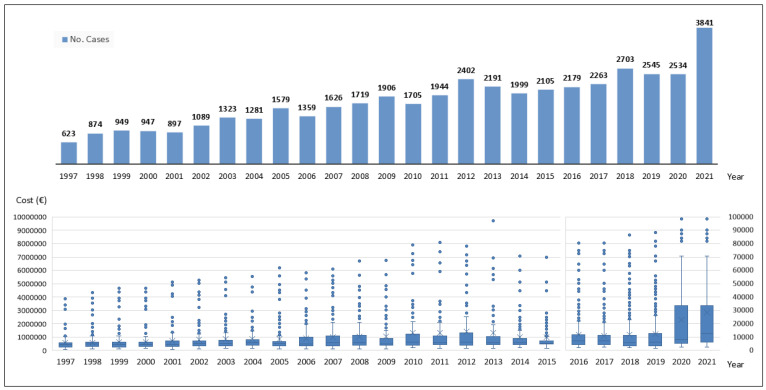
Evolution of caseload (bar chart) and cost (box plot) year by year, Spain (1997–2021).

**Figure 2 jof-10-00733-f002:**
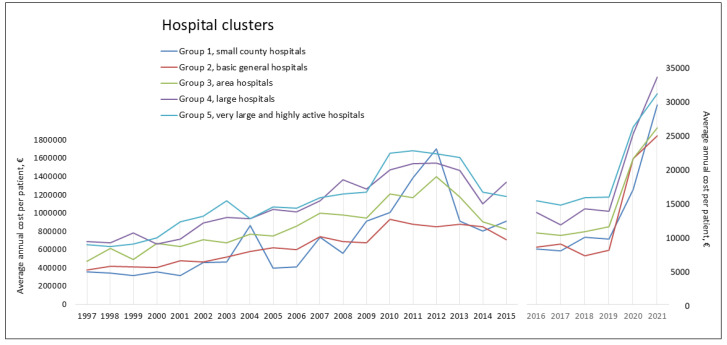
Evolution of the average annual cost per patient (EUR) and hospital clusters, Spain (1997–2021).

**Figure 3 jof-10-00733-f003:**
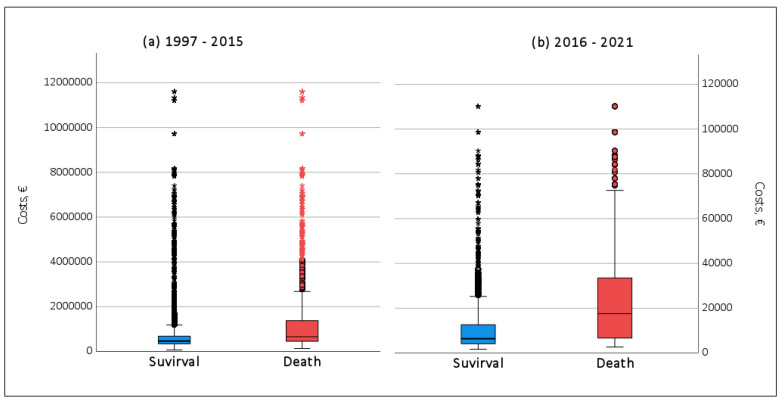
Box plot of the cost incurred in hospitalized patients with aspergillosis who survived and died in Spain over 25 years: (**a**) 1997–2015, (**b**) 2016–2021.

**Table 1 jof-10-00733-t001:** Characteristics of patients with aspergillosis-related hospitalizations, 1997–2021.

Variables		N = 44,583 Cases (100%)
Age (years)		n (%)
Pediatric population	0–14 years old	1347 (3.0)
Adult population	15–44 years old	5901 (13.2)
	45–64 years old	13,616 (30.5)
	65–74 years old	11,771 (26.4)
	≥75 years old	11,948 (26.8)
Mean (±SD)		61.6 (±18.762)
Gender		n (%)
Male		30,023 (67.3)
Female		14,560 (32.7)
Aspergillosis causing hospitalization		n (%)
Principal diagnosis		8111 (18.2)
Secondary diagnosis		36,472 (81.8)
Comorbidity		n (%)
Respiratory disease	Chronic obstructive pulmonary disease	19,630 (44.0)
	Influenza/Pneumonia	14,439 (32.4)
Oncological or malignant process	Hematology malignancy	9081 (20.4)
	Non-Hematology malignancy	6253 (14.0)
Admission type		n (%)
Urgent/Non-programmed		34,570 (77.5)
Programmed		9873 (22.1)
Unknown		140 (0.3)
Process type		n (%)
Medical		36,138 (81.1)
Surgical		8362 (18.8)
Unclassified		83 (0.2)
Hospital clusters		n (%)
GROUP 1: Small regional hospitals		1937 (4.3)
GROUP 2: Basic general hospitals		7370 (16.5)
GROUP 3: Area hospitals		11,459 (25.7)
GROUP 4: Large hospitals		9627 (21.6)
GROUP 5: Complex hospitals		11,864 (26.6)
Unclassifiable hospitals		2326 (5.2)
Hospital financing system		n (%)
Social insurance		43,246 (97.0)
Private insurance		342 (0.8)
Local corporations/Cabildos		211 (0.5)
Mutual health insurance companies		193 (0.4)
Mixed insurance		85 (0.2)
Workplace accidents		76 (0.2)
Traffic accidents		60 (0.1)
Unknown		370 (0.8)
Hospital service		n (%)
Internal medicine		9257 (20.8)
Pneumology		8791 (19.7)
Clinical hematology		6003 (13.5)
Special Units		n (%)
CCU/ICU		3512 (7.9)
Neonatal intensive care unit/Pediatric intensive care unit		68 (0.1)
Transplant unit		650 (1.5)
Hospital stay (days)		Mean (±SD)
Hospital		27 (±27.398)
CCU/ICU		25 (±26.172)
Circumstance to hospital discharge		n (%)
Home		30,511 (68.4)
Transferred to another hospital		1523 (3.4)
Transferred to social-health center		705 (1.6)
Voluntary discharge		127 (0.3)
Unknown		555 (1.2)
	ROM, n (%)	SOI, n (%)
0	6780 (15.2)	6780 (15.2)
1 = Minor	1563 (3.5)	344 (0.8)
2 = Moderate	8463 (19.0)	4069 (9.1)
3 = Major	15,137 (34.0)	15,963 (35.8)
4 = Extreme	12,640 (28.4)	17,427 (39.1)
Outcome		n (%)
Survival		33,421 (75.0)
Death		11,162 (25.0)

Abbreviations: SD, standard deviation; CCU, critical care unit; ICU, intensive care unit; SOI, severity of illness; ROM, risk of mortality.

**Table 2 jof-10-00733-t002:** Cost of aspergillosis-related hospitalizations, 1997–2021.

Variables	1997–2021		1997–2015				ANOVA	2016–2021				ANOVA
Age (Years)	No.	Total Cost, EUR	No.	Total Cost, EUR	Mean Cost per Patient (±SD), EUR		*p*-Value *	No.	Total Cost, EUR	Mean Cost per Patient (±SD), EUR		*p*-Value *
0–14 years old	1346	1,228,162,673	862	1,219,212,278	1,414,399	(±1,652,117)	<0.001	484	8,950,395	18,492	(±19,687)	<0.001
15–64 years old	19,506	15,823,788,660	12,922	15,688,025,505	1,214,055	(±1,624,955)		6584	135,763,154	20,620	(±24,846)	
≥65 years old	23,710	12,122,812,538	14,713	11,985,331,350	814,608	(±1,264,225)		8997	137,481,187	15,280	(±21,379)	
Gender												
Male	30,010	19,488,664,209	19,501	19,297,689,423	989,574	(±1,442,660)	<0.001	10,509	190,974,785	18,172	(±23,618)	<0.001
Female	14,556	9,686,099,663	8996	9,594,879,711	1,066,571	(±1,513,346)		5556	91,219,952	16,418	(±21,631)	
Diagnosis												
Principal diagnosis	8111	4,099,926,669	5781	4,079,448,656	705,664	(±804,518)	<0.001	2330	20,478,013	8788	(±8192)	<0.001
Secondary diagnosis	36,451	25,074,837,203	22,716	24,813,120,478	1,092,319	(±1,581,197)		13,735	261,716,724	19,054	(±24,294)	
Comorbidity												
COPD	19,630	8,817,805,808	12,638	8,741,401,412	691,676	(±1,060,876)	<0.001	6988	76,404,396	10,933	(±16,909)	<0.001
Influenza/Pneumonia	14,439	11,597,012,286	9351	11,456,413,100	1,225,153	(±1,592,216)		5086	140,599,185	27,644	(±29,250)	
Hematology malignancy	9081	8,868,538,550	6399	8,816,125,825	1,377,734	(±1,344,120)		2680	52,412,725	19,556	(±17,510)	
Non-Hematology malignancy	6253	3,397,598,143	3554	3,361,592,995	945,861	(±1,426,588)		2695	36,005,148	13,359	(±16,964)	
Admission type												
Urgent	34,554	20,335,082,541	21,617	20,111,268,135	930,345	(±1,401,719)	<0.001	12,937	223,814,405.97	17,300.33	(±23,216)	0.007
Programmed	9868	8,729,142,441	6803	8,672,337,661	1,274,781	(±1,620,189)		3065	56,804,779.51	18,533.37	(±21,715)	
Hospital service												
Internal medicine	9251	3,280,538,703	4963	3,222,458,726	649,296	(±1,012,201)	<0.001	4288	58,079,977.26	13,544.77	(±20,693) (20,693.31)	<0.001
Pneumology	8789	3,171,019,710.72	4832	3,128,539,728.00	647,462.69	(±1,038,943.42)		3957	42,479,982.72	10,735.40	(±18,029.39)	
Hematology	6000	5,802,240,016.24	3953	5,763,262,729.00	1,457,946.55	(±1,384,024.50)		2047	38,977,287.24	19,041.17	(±6765.31)	
Special Units												
CCU/ICU	3510	4,152,270,859.65	1585	4,078,186,645.00	2,572,988.42	(±2,683,430.06)	<0.001	1925	74,084,214.65	38,485.31	(±29,594.24)	<0.001
Transplant unit	650	718,263,569.88	381	712,970,022.00	1,871,312.39	(±2,448,535.57)		269	5,293,547.88	19,678.62	(±27,009.54)	
Process type												
Medical	36,138	14,926,137,615.05	23,501	14,808,999,245.39	630,143.37	(±602,512.98)	<0.001	12,637	117,138,369.66	9269.48	(±8370.94)	<0.001
Surgical	8362	14,235,509,249.17	4975	14,070,652,356.70	2,828,271.83	(±2,569,841.75)		3387	164,856,892.47	48,673.42	(±1840.10)	
Hospital clusters												
GROUP 1	1937	674,120,094.85	718	655,957,101.46	913,589.28	(±1,652,097.98)	<0.001	1219	18,162,993.38	14,899.91	(±21,896.75)	<0.001
GROUP 2	7366	3,450,112,914.36	4826	3,417,345,238.47	708,111.32	(±1,146,770.18)		2540	32,767,675.89	12,900.66	(±19,513.36)	
GROUP 3	11,455	7,073,487,370.15	7380	7,006,931,168.43	949,448.67	(±1,318,514.24)		4075	66,556,201.72	16,332.81	(±21,211.54)	
GROUP 4	9624	7,183,218,863.12	6156	7,112,623,811.92	1,155,396.98	(±1,607,202.84)		3468	70,595,051.20	20,356.13	(±24,732.31)	
GROUP 5	11,854	9,518,082,874.07	7717	9,431,816,051.41	1,222,212.78	(±1,676,358.52)		4137	86,266,822.65	20,852.51	(±25,232.56)	
Hospital financing system												
Social insurance	43,227	27,935,493,013.87	27,700	27,665,204,194.95	998,743.83	(±1,439,487.18)	<0.001	15,527	270,288,818.92	17,407.66	(±22,852.35)	<0.001
Private insurance	342	384,286,465.57	274	382,406,276.07	1,395,643.34	(±2,083,609.37)		68	1,880,189.50	27,649.84	(±29,290.55)	
Mutual insurance co.	193	191,094,277.02	128	189,332,569.44	1,479,160.70	(±2,014,685.46)		65	1,761,707.58	27,103.19	(±28,103.08)	
Local corporations	211	16,098,466.49	13	12,615,343.20	970,411.01	(±1,329,402.03)		198	3,483,123.28	17,591.53	(±23,850.65)	
Mixed insurance	85	3,401,898.02	2	2,068,965.85	1,034,482.92	(±365,630.67)		83	1,332,932.17	16,059.42	(±18,142.25)	
Traffic accidents	84	191,675,393.51	57	164,563,994.32	2,887,087.62	(±2,620,096.33)		3	138,656.33	46,218.78	(±25,015.41)	
Workplace accidents	52	67,667,854.95	37	93,046,092.00	2,514,759.24	(±2,530,412.20)		39	1,594,505.,80	40,884.76	(±230,119.77)	
SOI												
0	6759	4,688,886,922.01	6718	4,688,687,446.09	697,929.06	(±840,578.44)	<0.001	41	199,475.92	4865.27	(±382.84)	<0.001
1 = Minor	344	144,440,398.32	166	143,453,197.00	864,175.88	(±651,864.39)		178	987,201.32	5546.07	(±3179.70)	
2 = Moderate	4069	1,607,665,856.55	2213	1,597,977,325.00	722,086.45	(±1,062,270.38)		1856	9,688,531.55	5220.11	(±4622.62)	
3 = Major	15,963	6,741,727,004.81	9641	6,691,445,456.00	694,061.35	(±887,694.00)		6322	50,281,548.81	7953.42	(±9295.86)	
4 = Extreme	17,427	15,992,043,691.45	9759	15,771,005,711.00	1,616,047.31	(±2,047,940.47)		7668	221,037,980.45	28,826.03	(±28,004.46)	
ROM												
0	6759	4,688,886,922.01	6718	4,688,687,446.09	697,929.06	(±840,578.44)	<0.001	41	199,475.92	4865.27	(±382.84)	<0.001
1 = Minor	1563	500,921,137.70	717	495,828,610.00	691,532.23	(±673,750.59)		846	5,092,527.70	6019.54	(±4578.27)	
2 = Moderate	8463	3,986,521,165.86	5022	3,955,714,885.00	787,677.20	(±1,030,187.55)		3441	30,806,280.86	8952.71	(±11,163.21)	
3 = Major	15,137	8,419,052,695.23	8755	8,311,002,289.00	949,286.38	(±1,335,484.32)		6382	108,050,406.23	16,930.49	(±22,640.57)	
4 = Extreme	12,640	11,579,381,952.33	7285	11,441,335,905.00	1,570,533.41	(±2,093,144.78)		5355	138,046,047.33	25,778.91	(±27,389.04)	
Outcome												
Survival	33,404	18,732,760,191.76	21,340	18,562,960,680.10	869,866.95	(±1,258,363.43)	<0.001	12,064	169,799,511.66	14,074.89	(±19,959.76)	<0.001
Death	11,158	10,442,003,681.38	7157	10,329,608,454.99	1,443,287.47	(±1,893,976.31)		4001	112,395,226.40	28,091.78	(±27,730.75)	
Global cost	44,562	29,174,763,873.15	28,497	28,892,569,135.09	1,013,881.08	(±1,465,753.29)		16,065	282,194,738.06	17,565.81	(±22,965.11)	
Pneumology	8789	3,171,019,710	4832	3,128,539,728	647,462	(±1,038,943)		3957	42,479,982	10,735	(±18,029)	
Hematology	6000	5,802,240,016	3953	5,763,262,729	1,457,946	(±1,384,024)		2047	38,977,287	19,041	(±6765)	
Special units												
CCU/ICU	3510	4,152,270,859	1585	4,078,186,645	2,572,988	(±2,683,430)	<0.001	1925	74,084,214	38,485	(±29,594)	<0.001
Transplant unit	650	718,263,569	381	712,970,022	1,871,312	(±2,448,535)		269	5,293,547	19,678	(±27,009)	
Process type												
Medical	36,138	14,926,137,615	23,501	14,808,999,245	630,143	(±602,512)	<0.001	12,637	117,138,369	9269	(±8370)	<0.001
Surgical	8362	14,235,509,249	4975	14,070,652,356	2,828,271	(±2,569,841)		3387	164,856,892	48,673	(±1840)	
Hospital clusters												
GROUP 1	1937	674,120,094	718	655,957,101	913,589	(±1,652,097)	<0.001	1219	18,162,993	14,899	(±21,896)	<0.001
GROUP 2	7366	3,450,112,914	4826	3,417,345,238	708,111	(±1,146,770)		2540	32,767,675	12,900	(±19,513)	
GROUP 3	11,455	7,073,487,370	7380	7,006,931,168	949,448	(±1,318,514)		4075	66,556,201	16,332	(±21,211)	
GROUP 4	9624	7,183,218,863	6156	7,112,623,811	1,155,396	(±1,607,202)		3468	70,595,051	20,356	(±24,732)	
GROUP 5	11,854	9,518,082,874	7717	9,431,816,051	1,222,212	(±1,676,358)		4137	86,266,822	20,852	(±25,322)	
Hospital financing system												
Social insurance	43,227	27,935,493,013	27,700	27,665,204,194	998,743	(±1,439,487)	<0.001	15,527	270,288,818	17,407	(±22,852)	<0.001
Private insurance	342	384,286,465	274	382,406,276	1,395,643	(±2,083,609)		68	1,880,189	27,649	(±29,290)	
Mutual insurance co.	193	191,094,277	128	189,332,569	1,479,160	(±2,014,685)		65	1,761,707	27,103	(±28,103)	
Local corporations	211	16,098,466	13	12,615,343	970,411	(±1,329,402)		198	3,483,123	17,591	(±23,850)	
Mixed insurance	85	3,401,898	2	2,068,965	1,034,482	(±365,630)		83	1,332,932	16,059	(±18,142)	
Traffic accidents	84	191,675,393	57	164,563,994	2,887,087	(±2,620,096)		3	138,656	46,218	(±25,015)	
Workplace accidents	52	67,667,854	37	93,046,092	2,514,759	(±2,530,412)		39	1,594,505	40,884	(±230,119)	
SOI												
0	6759	4,688,886,922.01	6718	4,688,687,446.09	697,929.06	(±840,578.44)	<0.001	41	199,475.92	4865.27	(±382.84)	<0.001
1 = Minor	344	144,440,398.32	166	143,453,197.00	864,175.88	(±651,864.39)		178	987,201.32	5546.07	(±3179.70)	
2 = Moderate	4069	1,607,665,856.55	2213	1,597,977,325.00	722,086.45	(±1,062,270.38)		1856	9,688,531.55	5220.11	(±4622.62)	
3 = Major	15,963	6,741,727,004.81	9641	6,691,445,456.00	694,061.35	(±887,694.00)		6322	50,281,548.81	7953.42	(±9295.86)	
4 = Extreme	17,427	15,992,043,691.45	9759	15,771,005,711.00	1,616,047.31	(±2,047,940.47)		7668	221,037,980.45	28,826.03	(±28,004.46)	
ROM												
0	6759	4,688,886,922.01	6718	4,688,687,446.09	697,929.06	(±840,578.44)	<0.001	41	199,475.92	4865.27	(±382.84)	<0.001
1 = Minor	1563	500,921,137.70	717	495,828,610.00	691,532.23	(±673,750.59)		846	5,092,527.70	6019.54	(±4578.27)	
2 = Moderate	8463	3,986,521,165.86	5022	3,955,714,885.00	787,677.20	(±1,030,187.55)		3441	30,806,280.86	8952.71	(±11,163.21)	
3 = Major	15,137	8,419,052,695.23	8755	8,311,002,289.00	949,286.38	(±1,335,484.32)		6382	108,050,406.23	16,930.49	(±22,640.57)	
4 = Extreme	12,640	11,579,381,952.33	7285	11,441,335,905.00	1,570,533.41	(±2,093,144.78)		5355	138,046,047.33	25,778.91	(±27,389.04)	
Outcome												
Survival	33,404	18,732,760,191.76	21,340	18,562,960,680.10	869,866.95	(±1,258,363.43)	<0.001	12,064	169,799,511.66	14,074.89	(±19,959.76)	<0.001
Death	11,158	10,442,003,681.38	7157	10,329,608,454.99	1,443,287.47	(±1,893,976.31)		4001	112,395,226.40	28,091.78	(±27,730.75)	
Global cost	44,562	29,174,763,873.15	28,497	28,892,569,135.09	1,013,881.08	(±1,465,753.29)		16,065	282,194,738.06	17,565.81	(±22,965.11)	
Hospital financing system												
Social insurance	43,227	27,935,493,013	27,700	27,665,204,194	998,743	(±1,439,487)	<0.001	15,527	270,288,818	17,407	(±22,852)	<0.001
Private insurance	342	384,286,465	274	382,406,276	1,395,643	(±2,083,609)		68	1,880,189	27,649	(±29,290)	
Mutual insurance co.	193	191,094,277	128	189,332,569	1,479,160	(±2,014,685)		65	1,761,707	27,103	(±28,103)	
Local corporations	211	16,098,466	13	12,615,343	970,411	(±1,329,402)		198	3,483,123	17,591	(±23,850)	
Mixed insurance	85	3,401,898	2	2,068,965	1,034,482	(±365,630)		83	1,332,932	16,059	(±18,142)	
Traffic accidents	84	191,675,393	57	164,563,994	2,887,087	(±2,620,096)		3	138,656	46,218	(±25,015)	
Workplace accidents	52	67,667,854	37	93,046,092	2,514,759	(±2,530,412)		39	1,594,505	40,884.	(±230,119)	
SOI												
0	6759	4,688,886,922	6718	4,688,687,446	697,929	(±840,578)	<0.001	41	199,475	4865	(±382)	<0.001
1 = Minor	344	144,440,398	166	143,453,197	864,175	(±651,864)		178	987,201	5546	(±3179)	
2 = Moderate	4069	1,607,665,856	2213	1,597,977,325	722,086	(±1,062,270)		1856	9,688,531	5220	(±4622)	
3 = Major	15,963	6,741,727,004	9641	6,691,445,456	694,061	(±887,694)		6322	50,281,548	7953	(±9295)	
4 = Extreme	17,427	15,992,043,691	9759	15,771,005,711	1,616,047	(±2,047,940)		7668	221,037,980	28,826	(±28,004)	
ROM												
0	6759	4,688,886,922	6718	4,688,687,446	697,929	(±840,578)	<0.001	41	199,475	4865	(±382)	<0.001
1 = Minor	1563	500,921,137	717	495,828,610	691,532	(±673,750)		846	5,092,527	6019	(±4578)	
2 = Moderate	8463	3,986,521,165	5022	3,955,714,885	787,677	(±1,030,187)		3441	30,806,280	8952	(±11,163)	
3 = Major	15,137	8,419,052,695	8755	8,311,002,289	949,286	(±1,335,484)		6382	108,050,406	16,930	(±22,640)	
4 = Extreme	12,640	11,579,381,952	7285	11,441,335,905	1,570,533	(±2,093,144)		5355	138,046,047	25,778	(±27,389)	
Outcome												
Survival	33,404	18,732,760,191	21,340	18,562,960,680	869,866	(±1,258,363)	<0.001	12,064	169,799,511	14,074	(±19,959)	<0.001
Death	11,158	10,442,003,681	7157	10,329,608,454	1,443,287	(±1,893,976)		4001	112,395,226	28,091	(±27,730)	
Global cost	44,562	29,174,763,873	28,497	28,892,569,135	1,013,881	(±1,465,753)		16,065	282,194,738	17,565	(±22,965)	
ROM												
0	6759	4,688,886,922	6718	4,688,687,446	697,929	(±840,578)	<0.001	41	199,475	4865	(±382.84)	<0.001
1 = Minor	1563	500,921,137	717	495,828,610	691,532	(±673,750)		846	5,092,527	6019	(±4578.27)	
2 = Moderate	8463	3,986,521,165	5022	3,955,714,885	787,677	(±1,030,187)		3441	30,806,280	8952	(±11,163.21)	
3 = Major	15,137	8,419,052,695	8755	8,311,002,289	949,286	(±1,335,484)		6382	108,050,406	16,930	(±22,640.57)	
4 = Extreme	12,640	11,579,381,952	7285	11,441,335,905	1,570,533	(±2,093,144)		5355	138,046,047	25,778	(±27,389.04)	
Outcome												
Survival	33,404	18,732,760,191	21,340	18,562,960,680	869,866	(±1,258,363)	<0.001	12,064	169,799,511	14,074	(±19,959)	<0.001
Death	11,158	10,442,003,681	7157	10,329,608,454	1,443,287	(±1,893,976)		4001	112,395,226	28,091	(±27,730)	
Global cost	44,562	29,174,763,873	28,497	28,892,569,135	1,013,881	(±1,465,753)		16,065	282,194,738	17,565	(±22,965)	

* Statistical significance level of 5% (*p* < 0.05). Abbreviations: SD, standard deviation; COPD, chronic obstructive pulmonary disease; CCU, Critical Care Unit; ICU, Intensive Care Unit; SOI, Severity of illness; ROM, Risk of mortality.

## Data Availability

The necessary information was obtained from the Minimum Basic Data Set (MBDS/CMBD in Spanish), from its first to its latest version, which is called the Register of Specialized Health Care (RAE-CMBD in Spanish).

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
