# Peer review of "Healthcare Costs of Hospitalizations Due to Aspergillosis and 25-Year Trends in Spain, 1997–2021"

_jof, 2024, doi:10.3390/jof10110733_

Round 1
Reviewer 1 Report
This is an important study detailing the number of documented IA cases in Spain over 24 years and the costs of managing those cases. The number of cases rose substantially in the first years and again in 2021. Some of these changes are only superficially explored in the manuscript.
The relatively high number of cases in neonates is remarkable; the relatively even number of cases in different age brackets in adults equally interesting and might be used to estimate YLL from IA in the final 5 years..
Comments:
The abstract is missing some key actual costs. Provide some key costs in the abstract, such as the mean and range 2021 per patient costs.
L62 – COPD, influenza, Covid-19 – also major contributors to additional cases.
L151 If a patient developed IA during an admission for another condition such as leukaemia, was this included, or did the admission have to be mainly as a result of IA? This needs clarification.
L152 Were second admissions related to IA removed, or added to the original cost?
L187. What proportion od patients are treated in the private sector? Would the authors argue that >95% of patients with IA are managed in the public sector because of the complexity of illness? Table 1 lists various insurance plans.
L195. The authors use Aspergillus infection here and so was chronic pulmonary aspergillosis excluded or included; see DOI: 10.1111/myc.12950 as an example.
L205 and Table 1. Presumably these are in hospital deaths only, not the total mortality at 30 day, 6 weeks or 12 weeks as usually reported in clinical trials of IA. Please confirm.
Figure 1 is confusing. There are no units on the vertical access. The numbers could be numbers of patients (blue bars) or costs, and if costs are they total national costs or individual patients costs – the right hand panel looks like individual costs. Suggest the number of patients is put on a right hand side vertical access and all costs are displayed by patient, not totals, with a left hand side labelled vertical axis.
Figure 2 has a similar confusion of total or individual costs. Be consistent within the same figure.
Table 2. Remove the 2 decimal points for all costs – not necessary and implies a precision that is unrealistic.
Table 2. Repeat the column headers on each page of the table.
L221. Add some text here. Put the main mean (range) costs per patients in the top categories into the text, ie ICU. COPD, Influenza, Haematology, NICU. Do the same for the deaths vs survivors. The table is tough to read and the text needs the highlights.
Figure 3 vertical axes are unreadable and probably confusing. Bottom right might read 200CC.00, but whatever it is, is unclear. 2 decimal points are not required for euros, if that is what is meant by the .00.
The jump in costs and number of cases in 2021 is unexplained. Some factors leading to this significant increase should be added.
The gradual increase in numbers of cases from 1997 – 2009 remains unexplained. Do the authors infer than many cases of IA were missed before this? See Danion doi: 10.1093/mmy/myy081. Has the introduction of galactomannan been instrumental in an increase of cases? Or was it the awareness about COPD as a key underlying disease been important? See Guinea DOI 10.1111/j.1469-0691.2009.03015.x
No data is provided on inflation-adjusted costs. Using a per-patient cost and simple histogram of costs adjusted by inflation per year would be a helpful additional figure.
L374-8. Not clear what the authors are saying, and if they used the actual mean costs here, would make it clearer.
L390 Providing data on standard daily costs of different drugs might be helpful here. Especially ambisome versus the azoles voriconazole and isavuconazole.
Minor comments
Italicise Aspergillus and its species such as A. terreus
L119 what is and MIR doctor?
L181, L223 either aspergillosis or Aspergillus infection, but not Aspergillus alone.
Table 1. Pneumology or Respiratory medicine
Table 1. Legend – aspergillosis-related. Not Aspergillosis
L360 What is NAC?
L364 syntax of ‘driver’ not right
Author Response
For research article
|
Response to Reviewer Comments |
|||
|
1. Summary |
|
|
|
|
Thank you very much for taking the time to review this manuscript. Please find the detailed responses below and the corresponding corrections highlighted in the re-submitted files.
|
|||
|
|
|
||
|
|
|
||
|
2. Point-by-point response to Comments and Suggestions for Authors |
|||
|
Comments 1: The abstract is missing some key actual costs. Provide some key costs in the abstract, such as the mean and range 2021 per patient costs. Response 1: DONE L38-40
Comments 2: L62 – COPD, influenza, Covid-19 – also major contributors to additional cases. Response 2: Of course, we mention it in the line 436.
Comments 3 L151 If a patient developed IA during an admission for another condition such as leukaemia, was this included, or did the admission have to be mainly as a result of IA? This needs clarification. Response 3: As we specified in the line 154, hospitalized cases with IA were collected as the principal or secondary diagnosis (in many of these the cause of admission was not IA, and it could present throughout the admission).
Comments 4 L152 Were second admissions related to IA removed, or added to the original cost? Response 4: Every income is a cost.
Comments 5 L187. What proportion od patients are treated in the private sector? Would the authors argue that >95% of patients with IA are managed in the public sector because of the complexity of illness? Table 1 lists various insurance plans. Response 5: As we specified in the line 489, ¨this study only considered cases in public hospitals and not non-hospital cases or those in private centers”. The Spanish healthcare model provides greater public coverage, not only due to complexity but also due to capacity.
Comments 6 L195. The authors use Aspergillus infection here and so was chronic pulmonary aspergillosis excluded or included; see DOI: 10.1111/myc.12950 as an example. Response 6: it´s included. DONE in L 156.
Comments 7 L205 and Table 1. Presumably these are in hospital deaths only, not the total mortality at 30 day, 6 weeks or 12 weeks as usually reported in clinical trials of IA. Please confirm. Response 7: Yes, in hospital deaths
Comments 8 Figure 1 is confusing. There are no units on the vertical access. The numbers could be numbers of patients (blue bars) or costs, and if costs are they total national costs or individual patients costs – the right hand panel looks like individual costs. Suggest the number of patients is put on a right hand side vertical access and all costs are displayed by patient, not totals, with a left hand side labelled vertical axis. Response 8: DONE. We have replaced the figure with a box diagram.
Comments 9 Figure 2 has a similar confusion of total or individual costs. Be consistent within the same figure. Response 9: DONE.
Comments 10 Table 2. Remove the 2 decimal points for all costs – not necessary and implies a precision that is unrealistic. Response 10: DONE
Comments 11 Table 2. Repeat the column headers on each page of the table. Response 11: DONE
Comments 12 L221. Add some text here. Put the main mean (range) costs per patients in the top categories into the text, ie ICU. COPD, Influenza, Hematology, NICU. Do the same for the deaths vs survivors. The table is tough to read and the text needs the highlights. Response 12: DONE
Comments 13 Figure 3 vertical axes are unreadable and probably confusing. Bottom right might read 200CC.00, but whatever it is, is unclear. 2 decimal points are not required for euros, if that is what is meant by the .00. Response 13: DONE.
Comments 14 The jump in costs and number of cases in 2021 is unexplained. Some factors leading to this significant increase should be added. Response 14: it´s explicated in line 434. ¨The second peak between 2020 and 2021 coincided with the SARS-CoV-2 pandemic [2,17]¨
Comments 15 The gradual increase in numbers of cases from 1997 – 2009 remains unexplained. Do the authors infer than many cases of IA were missed before this? See Danion doi: 10.1093/mmy/myy081. Has the introduction of galactomannan been instrumental in an increase of cases? Or was it the awareness about COPD as a key underlying disease been important? See Guinea DOI 10.1111/j.1469-0691.2009.03015.x Response 15: In the last two decades, an increase in the incidence of invasive pulmonary aspergillosis has been observed in critically ill patients with hemopathies, transplants or neutropenia, but each increasingly associated with other diseases such as COPD. In fact, in patients with COPD, mortality is higher than in hematological patients (89% versus 59%), partly due to the delay in diagnosis and treatment. (BB: García-Aguinaga ML, Rodríguez-González CT, Sabado-Angngasing EJ, Velayos-Rubio R, González-González J. Invasive pulmonary Aspergyllosis in COPD patient. Rev Esp Casos Clin Med Intern (RECCMI). 2020 (Abr); 5(1): 33-35. doi: 10.32818/reccmi.a5n1a12.)
Comments 16 No data is provided on inflation-adjusted costs. Using a per-patient cost and simple histogram of costs adjusted by inflation per year would be a helpful additional figure Response 16: The weights and costs of hospitalization processes provided by the CMBD serve as standards of cost results in the field of health management, so our objective was to provide descriptive clinical-administrative information, without pretensions of doing an exhaustive economic analysis.
Comments 17 L374-8. Not clear what the authors are saying, and if they used the actual mean costs here, would make it clearer. Response 18: DONE
Comments 18 L390 Providing data on standard daily costs of different drugs might be helpful here. Especially ambisome versus the azoles voriconazole and isavuconazole.
Comments 19 Italicise Aspergillus and its species such as A. terreus Response 19: DONE
Comments 20 L119 what is and MIR doctor? Response 20: Medical Intern Resident: a doctor interning at a hospital or health centre to obtain the diploma of specialist in a branch of medicine in Spain. DONE
Comments 21 L181, L223 either aspergillosis or Aspergillus infection, but not Aspergillus alone. Response 21: On line 181 that expression does not appear. L223 done.
Comments 22 Table 1. Pneumology or Respiratory medicine Response 22: Pneumology
Comments 23 Table 1. Legend – aspergillosis-related. Not Aspergillosis Response 23: I don´t understand
Comments 24 L360 What is NAC? Response 24: Community-acquired pneumonia (CAP) DONE
Comments 25 L364 syntax of ‘driver’ not right Response 25: DONE
|
|||
Reviewer 2 Report
This is a descriptive study reviewing the cost of hospitalization with Aspergillosis over a 25-year period in Spain. As one might predict the cost has risen significantly over 25 years. It would be helpful to the reader to understand if whether this rise mirrors the rate of inflation in Spain over this time period and if it diverges from the rise in cost seen for other infections over this time period in Spain. Some discussion of similar reports from other countries might help put the increase cost due to Aspergillosis in perspective relative to other rising costs over this same time period.
Table 1 Hospital Service section - should it read "Neurology" instead of "Neumology"? Same question in Table 2 under the Hospital Service section.
Page 13 lin3 390 - please indicate what year isavuconazole was introduced in Spain.
Author Response
For research article
|
Response to Reviewer Comments |
|||
|
1. Summary |
|
|
|
|
Thank you very much for taking the time to review this manuscript. Please find the detailed responses below and the corresponding corrections highlighted in the re-submitted files.
|
|||
|
|
|
||
|
|
|
||
|
2. Point-by-point response to Comments and Suggestions for Authors |
|||
Comments 1 This is a descriptive study reviewing the cost of hospitalization with Aspergillosis over a 25-year period in Spain. As one might predict the cost has risen significantly over 25 years. It would be helpful to the reader to understand if whether this rise mirrors the rate of inflation in Spain over this time period and if it diverges from the rise in cost seen for other infections over this time period in Spain. Some discussion of similar reports from other countries might help put the increase cost due to Aspergillosis in perspective relative to other rising costs over this same time period.
Response 1: Our objective was to provide descriptive clinical-administrative information, without pretensions of making an exhaustive economic analysis.
Comments 2 Table 1 Hospital Service section - should it read "Neurology" instead of "Neumology"? Same question in Table 2 under the Hospital Service section.
Response 2: DONE
Comments 3 Page 13 line 390 - please indicate what year isavuconazole was introduced in Spain.
Response 3: DONE